# Plasma Kinetics of Choline and Choline Metabolites After A Single Dose of Superba*Boost*^TM^ Krill Oil or Choline Bitartrate in Healthy Volunteers

**DOI:** 10.3390/nu11102548

**Published:** 2019-10-22

**Authors:** Yvonne Mödinger, Christiane Schön, Manfred Wilhelm, Petter-Arnt Hals

**Affiliations:** 1BioTeSys GmbH, Schelztorstraße 54–56, 73728 Esslingen, Germany; 2Department of Mathematics, Natural and Economic Sciences, Ulm University of Applied Sciences, Albert-Einstein-Allee 55, 89081 Ulm, Germany; 3Aker BioMarine Antarctic AS, Oksenøyveien 10, 1327 Lysaker, Norway

**Keywords:** choline, pharmacokinetics, Antarctic krill, phosphatidylcholine, Superba*Boost*, choline bitartrate, betaine, dimethylglycine, trimethylamine N-oxide

## Abstract

As an essential nutrient, the organic water-soluble compound choline is important for human health. Choline is required for numerous biological processes, including the synthesis of neurotransmitters, and it is an important prerequisite for structural integrity and the functioning of cells. A choline-rich diet provides crucial choline sources, yet additional choline dietary supplements might be needed to fully meet the body’s requirements. Dependent on the structure of choline in different sources, absorption and metabolism may differ and strongly impact the bioavailability of circulating choline. This study in healthy volunteers aimed to compare the pharmacokinetics of free choline and of selected choline metabolites between the single dose intake of phosphatidylcholine, present in Superba*Boost*^TM^ krill oil, and choline bitartrate salt. Results demonstrate that albeit free choline levels in plasma were comparable between both choline sources, peak choline concentration was reached significantly later upon intake of Superba*Boost*^TM^. Moreover, the occurrence of choline metabolites differed between the study products. Levels of the biologically important metabolites betaine and dimethylglycine (DMG) were higher, while levels of trimethylamine N-oxide (TMAO) were substantially lower upon intake of Superba*Boost*^TM^ compared to choline bitartrate.

## 1. Introduction

Choline is an essential water-soluble nutrient, which can be formed by de novo synthesis but needs to be additionally obtained via the diet to meet the body’s requirements [1]. Choline and its metabolites play a key role in various mammalian biological processes [2]. They serve as components of structural lipoproteins and membrane lipids and thus promote structural integrity and the signaling functions of cells. Furthermore, choline serves as a precursor of the neurotransmitter acetylcholine, which is crucial for brain function [3]. Choline was officially recognized as an essential nutrient by the Institute of Medicine (IOM) in 1998.

Choline absorption and metabolism in the gut strongly depends on its biochemical formulation, as well as on the present gut microbiota [4]. An estimated 90% of the US population is not meeting the choline intake recommendations of 550 mg/day for men and 425 mg/day for women, as issued by the Institute of Medicine [1,5,6]. More recently, the European Food and Safety Authority (EFSA) recommended a slightly lower adequate intake (AI) of 400 mg choline per day for all adults. Still, the average choline intake of healthy adults in the EU (370 mg/day) remains below this level [7]. Low intake levels of choline may lead to fatigue and muscle dysfunction and can impair brain function and memory. In the long term, lacking choline can increase the risk of neurological disorders like Alzheimer’s disease [8,9]. In severe cases, choline deficiency promotes kidney necrosis and fatty liver disease [10,11], and a deficiency in phosphatidylcholine in the gut mucus layer is discussed as a causative factor for ulcerative colitis [12]. Furthermore, findings in rodents show that choline deficiency can promote heart dysfunction [13]. Also, pregnant women should ensure adequate levels of choline due to its importance in fetal development [14].

Thus, suitable choline dietary supplements are required where choline is well absorbed and is available in the circulation in meaningful amounts. Krill oil is an acknowledged dietary supplement to provide omega-3 fatty acids, but it is increasingly recognized to also be a useful source of the phospholipid phosphatidylcholine, and thus also for choline [15,16]. Superba*Boost*^TM^ krill oil is the major marine oil on the market with high concentrations of phosphatidylcholine. Superba*Boost*^TM^ contains particularly high levels of phospholipids (≥56%), out of which phosphatidylcholine makes up the major share (approximately 90%). In the intestinal tract, phosphatidylcholine may either be absorbed intact or after hydrolysis by phospholipase D, by which free choline is liberated from the structure [17]. In contrast, free choline is immediately available from the commonly supplemented choline salts, such as choline bitartrate or choline chloride, without the need for an enzymatic conversion step.

Besides being available as free choline for various requirements of the body, choline from choline supplements is further converted into different metabolites. Betaine is an oxidation product of free choline and is important, for the formation of the essential amino acid methionine, among others [18]. Dimethylglycine (DMG) is a metabolite of betaine and serves as an important source of the amino acid glycine and of methyl groups for biochemical reactions [19]. Trimethylamine N-oxide (TMAO) is metabolized in the host liver from trimethylamine (TMA) [20], which is, in turn, primarily generated from dietary choline, betaine, l-carnitine, and its metabolite, y-butyrobetaine, by the action of gut microbiota [21,22,23,24]. Recent findings implicate that TMAO is not merely a non-functional waste product of choline metabolism, but emerges as a risk factor and prognostic marker in the development and outcome of many diseases and disorders, including renal disease [23], cardiovascular disease [22,25,26,27,28,29], colorectal cancer [30], type II diabetes and neurological disorders [31]. Moreover, high TMAO levels seem to be related to inflammation, for instance by activating the NLRP3 inflammasome [32,33,34], or by correlating with elevated plasma tumor necrosis factor (TNF)-α levels in an otherwise healthy population [35].

Circulating TMAO levels show high inter- and intra-individual variation and are determined by a number of influencing factors [6,31,36]. As such, TMAO levels appear to increase with age [37,38], and are influenced by kidney function, especially by the activity of hepatic flavin monooxygenases. Of note, the diet also plays a key role in TMAO formation by impacting the gut microbial flora [4]. Western or high fat diets where shown to increase plasma TMAO levels [6,31,39,40], while there is reduced TMA production from l-carnitine in vegetarians [41]. Furthermore, intake of antibiotics, especially broad-spectrum antibiotics, can almost completely suppress TMA and TMAO production, but levels return back to normal within one month after withdrawal of the antibiotic [22,27].

Overall, bioavailability data of choline from different sources are rather limited and do not provide a full picture, including further metabolites which can exert independent physiologic effects. In this context, TMAO formation in particular is evolving as a topic of high interest. Previous studies have already demonstrated that phosphatidylcholine is only limitedly converted to undesirable trimethylamines in contrast to choline salts [21,42,43]; however, less is known regarding the other metabolites. Further, these studies did not investigate TMAO formation after phosphatidylcholine intake in form of krill oil, or, in particular, in the form of Superba*Boost*^TM^. Since Superba*Boost*^TM^ is discussed as a suitable choline source, circulating levels of choline and choline metabolites, especially TMAO, remain to be investigated.

Thus, the aim of the current study was to investigate the kinetics of choline and the generation of choline metabolites from two different choline sources. Concentration and plasma kinetics of choline and its metabolites betaine, DMG and TMAO were investigated after intake of Superba*Boost*^TM^ krill oil and choline bitartrate.

The study was performed in a randomized cross-over design with an additional control group receiving fish oil containing comparable amounts of the omega-3 fatty acids eicosapentaenoic acid (EPA) and docosahexaenoic acid (DHA) present in the Superba*Boost*^TM^ krill oil product, but no choline. In fish oil, however, EPA and DHA are bound to triglycerides, while they are mainly bound to phospholipids in krill oil. The fish oil control group allowed the estimation of potential confounding factors by standardized meals or by other regulatory mechanisms such as the circadian rhythm. In a study by Park et al., endogenous choline levels increased during the daytime in between meals, but not immediately after food intake, suggesting that additional mechanisms like the circadian rhythm may influence circulating choline levels [44].

## 2. Materials and Methods

### 2.1. Study Subjects

The study cohort consisted of 18 healthy males and females (1:1), where 12 subjects received the study products and 6 subjects received the control product (Figure 1). All included subjects were recruited between October and November 2018. Overall, 24 subjects, aged 18–65 years with a body mass index (BMI) of 18–30 kg/m^2^, were screened for eligibility. Subjects were excluded from study participation in case of a relevant history of any medical disorder potentially interfering with the study (e.g., malabsorption, chronic gastro-intestinal diseases, heavy depression, diabetes, etc.), in case of a regular intake of drugs or dietary supplements interfering with the study (e.g., omega-3 fatty acids, krill oil, etc.), in case of drug, alcohol, or medication abuse, or in case of pregnancy and breastfeeding. In addition, reasons for non-inclusion were low hemoglobin, platelet and leukocyte levels, as well as markers of liver damage exceeding the normal limit unless related to primary disease. Study participants had a low habitual consumption of fatty fish and seafood, not exceeding a frequency of twice per month. Medications for the treatment of chronic diseases that did not affect the metabolism of the study product were permitted.

Subjects gave written informed consent to participate in the study prior to screening evaluations. Ethical approval was obtained from the Institutional Review Board (IRB) of Landesärztekammer Baden-Württemberg with the approval code F-2018-076. The study was conducted in accordance with the guidelines set forth by the International Council for Harmonisation of Technical Requirements for Pharmaceuticals for Human Use (ICH), the guidelines for Good Clinical Practice (GCP), and the Declaration of Helsinki regarding the treatment of human subjects in a study. The present study was registered with the German Clinical Trials Register (DRKS; ID: DRKS00015828).

### 2.2. Study Design

The study was performed as a randomized, placebo-controlled, monocentric, two-way cross-over pharmacokinetic study. In total, 18 healthy volunteers took part in the study, where 12 subjects received the study products and 6 subjects received the placebo control product. The study was conducted at the study site of BioTeSys GmbH (Esslingen, Germany). Blood sampling was performed at two kinetic intervention days, over a time period of 24 h each, after a single dose of the study products. There was a wash-out phase of 14 days between the kinetic days. Blood samples were taken before (0 h), and 0.5 h, 1 h, 2 h, 4 h, 8 h, 12 h and 24 h after intake of the study products to obtain concentration-time profiles of free choline and its metabolites in plasma. During the kinetic days, fluid and calorie intake was standardized and meals with low choline content were served. In addition, subjects were asked to adhere to a diet poor in choline three days prior to the kinetic days and to consume a standardized dinner on the evening before the kinetic days.

### 2.3. Intervention

The investigational product was Superba*Boost*^TM^ krill oil (#1198530200, Aker BioMarine Antarctic AS, Lysaker, Norway), a choline source with phosphatidylcholine. The product was administrated orally as a single dose provided in softgel capsules containing in total 8 g Superba*Boost*^TM^ krill oil, equivalent to 572 mg choline. Choline bitartrate (#CHB:C18060079, ZeinPharma®, Nauheim, Germany) was used as a reference product that was similarly administered orally as a single dose provided in capsules (vegetable fiber), containing a total of 620 mg choline according to an actual analysis certificate. Fish oil (#L14006818:12, Omega-3 Seefischöl 100, Doppelherz®, Queisser Pharma, Flensburg, Germany) was used as placebo. Six grams of fish oil contained 1800 mg of the omega-3 fatty acids EPA and DHA, a dose comparable to the EPA and DHA content (1840 mg) in 8 g of Superba*Boost*^TM^ krill oil. The intake of study products occurred under supervision of the study personnel. Manufacturing of the study products was carried out in compliance with Good Manufacturing Practice (GMP) conditions and all ingredients and capsules were of food grade quality and met European food regulations. Superba*Boost*^TM^ krill oil capsules were provided by Aker BioMarine. Capsules containing choline bitartrate or fish oil were obtained commercially.

### 2.4. Sample Collection, Processing and Analysis

Venous blood samples were taken at screening and study visits at the study site of BioTeSys GmbH (Esslingen, Germany). Samples were analyzed the same day or the next day for safety and blood routine parameters (differentiated hemogram and clinical laboratory) at an accredited laboratory (Synlab Medizinisches Versorgungszentrum, Leinfelden-Echterdingen, Germany). Blood samples were centrifuged at 3000× *g* for 10 min at 4 °C. Plasma aliquots were taken and stored at −80 °C until analysis of choline and its metabolites, which was performed by HPLC-MS/MS at Bevital AS (Bevital AS, Bergen, Norway) in a high throughput, low-volume, multianalyte fashion, as described by Holm et al. [45].

### 2.5. Methods for Safety (Adverse Events, Concomitant Medication and Tolerability)

During the study intervention, the subjects documented any adverse events (AEs) and concomitant medication in diaries. Tolerability of the study products was assessed at the end of the kinetic days.

### 2.6. Data Analysis and Statistics

All statistical tests were performed two-sided. A *p*-value < 0.05 was considered as statistically significant. Non-normality was evaluated with the Shapiro–Wilk test using a significance level of 0.10. Plasma concentrations of the analytes and their increase over time were analyzed using ANOVA with repeated measures or Friedman test, if appropriate. To allow direct comparison of free choline from the different sources, Superba*Boost*^TM^ and choline bitartrate, data sets were dose adjusted for free choline, but not for the metabolites betaine, DMG and TMAO. This takes into account that the choline content of the Superba*Boost*^TM^ product was 8.4% lower than the choline bitartrate product. All concentration–time curves show baseline-corrected values. Pharmacokinetic parameters were individually calculated based on the plasma concentration–time curves. The area under the observed concentration–time curve (AUC) above baseline, within 12 h (AUC_0-12 h_) and within 24 h (AUC_0–24 h_) were calculated by applying the trapezoidal rule with the *y*-axis defined by choline plasma concentration, and the *x*-axis defined by sampling time points. After log transformation, AUC_0-12 h_, AUC_0–24 h_ and peak concentration (C_max_) were evaluated using a linear mixed model, taking into account sequence, period, and product. Differences between the time to reach maximum concentration (T_max_) were evaluated by the Wilcoxon rank sum test using the intra-individual differences between the outcomes in both periods as the raw data with consideration of the cross-over analysis by strict separation of treatment effects from period effects.

Descriptive comparison of fish oil uptake pattern with the uptake pattern from Superba*Boost*^TM^ or choline bitartrate were evaluated by pairwise comparison using Student t-test or Wilcoxon rank sum test, if appropriate. Statistical evaluation, summary tables and graphs were generated using GraphPad Prism software (GraphPad Software, La Jolla, CA, USA) and SAS v9.3 statistical software (SAS Institute, Cary, NC, USA).

## 3. Results

### 3.1. Subject Characteristics

Descriptive demographic data of the study cohort are summarized in Table 1. Briefly, the study cohort, receiving either Superba*Boost*^TM^ or choline bitartrate and consisting of a total of 12 subjects, was a healthy, non-smoking study group with an average age of 43.3 years (95% confidence interval (CI): 34.0–52.6) and a BMI of 23.6 kg/m^2^ (95% CI: 21.8–25.3) (Table 1). Vital signs, blood routine parameters and baseline levels of choline and choline metabolites were within normal range. Nine out of 12 subjects (75%) reported to be regularly physically active; two of the subjects followed a vegetarian diet, while no study subjects were vegans. Subjects reported to either never consume fish (*n* = 1), to consume fish less than once per month (*n* = 8), or to consume fish once or twice per month (*n* = 3).

### 3.2. Choline Pharmacokinetics Upon Intake of The Single-Dosed Choline Sources Phosphatidylcholine and Choline Bitartrate

To investigate the plasma concentration of choline and its metabolites betaine, DMG and TMAO over time, blood measurements were performed at 0 h, 0.5 h, 1 h, 2 h, 4 h, 8 h, 12 h and 24 h post-dosing. Choline concentrations in plasma upon intake of choline bitartrate, Superba*Boost*^TM^ and fish oil are shown in Figure 2. Plasma concentrations of free choline were baseline-corrected and linearly dose-adjusted to allow for direct comparison of pharmacokinetic endpoints. Both study products showed a comparable uptake profile of choline after single dose intake, however, with on average slightly higher choline levels and a faster metabolism upon Superba*Boost*^TM^ intake in comparison to choline bitartrate. Both study products resulted in a significant increase of choline concentration up to the 8 h time point (*p* < 0.0001). After C_max_ was reached, choline levels declined steadily and significantly in the choline bitartrate and Superba*Boost*^TM^ groups, resulting in below baseline levels 24 h post-dosing. No increase in choline levels was observed in the fish oil group, except for a slight elevation between the 4 h and 8 h time points, followed by a steady decline to below-baseline levels after 24 h, comparable to choline levels after intake of the study products.

Pharmacokinetic parameters of choline upon intake of the study and control products are summarized in Table 2. AUC_0–24 h_, AUC_0–12 h_ and C_max_ of free choline were comparable between choline bitartrate and Superba*Boost*^TM^, but significantly higher compared to fish oil. T_max_ was significantly longer upon intake of Superba*Boost*^TM^ in comparison to choline bitartrate.

### 3.3. Pharmacokinetic Characteristics of the Choline Metabolites Betaine, Dimethylglycine (DMG) and Trimethylamine Oxide (TMAO) Upon Choline Intake

Betaine is an important derivate of choline. Pharmacokinetic evaluations of betaine therefore allow for detailed metabolic characterizations of different choline sources and choline dietary supplements. Betaine plasma concentrations upon intake of the study products are shown in Figure 3. All values were baseline-corrected. A significant betaine increase over time was observed in all study groups (*p* < 0.0001). The betaine concentration–time curve was slightly higher when choline was provided as phosphatidylcholine in Superba*Boost*^TM^ compared to choline bitartrate, and both curves were significantly elevated compared to the fish oil group.

Pharmacokinetic parameters of betaine upon intake of the study and control products are shown in Table 3. AUC_0–24 h_ and AUC_0–12 h_ of free betaine were slightly higher, and C_max_ was significantly higher upon intake of Superba*Boost*^TM^ compared to choline bitartrate. Additionally, C_max_ levels of betaine in both study groups were significantly elevated compared to those in the fish oil placebo group. No difference was observed regarding the T_max_ of betaine, neither between both study products, nor between the study and placebo products.

DMG is a choline metabolite that is generated from betaine by donation of a methyl group to homocysteine. Thus, this metabolite can serve as a measure for the contribution of choline to one-carbon metabolism, a key biochemical pathway that provides the carbon units required for critical cellular processes. Highly significant increases of DMG were observed upon intake of both Superba*Boost*^TM^ (*p* < 0.0001) and choline bitartrate (*p* < 0.0001), while this was not the case for fish oil (*p* = 0.1091), as shown in Figure 4. All values were baseline-corrected. Interestingly, in some subjects, DMG levels were highest 24 h after product intake.

AUC_0–12 h_ and AUC_0–24 h_ of DMG were increased by trend upon intake of Superba*Boost*^TM^ compared to choline bitartrate (Table 4). Correspondingly, C_max_ was also increased upon intake of Superba*Boost*^TM^ compared to choline bitartrate, and values of both study products were significantly higher compared to fish oil. No differences were observed regarding T_max_ levels of DMG, neither between the two choline sources, nor in comparison to the fish oil group.

The gut flora-dependent metabolite TMAO is formed from orally ingested choline and betaine. As its generation is associated with adverse cardiac and renal outcomes, among others, TMAO pharmacokinetic analyses are important measures to characterize choline metabolism from choline sources. TMAO plasma concentrations upon intake of the study products are shown in Figure 5. All values were baseline-corrected. There was a significant increase in TMAO levels after intake of both Superba*Boost*^TM^ (*p* < 0.0001) and choline bitartrate (*p* < 0.0001), although TMAO was increased to a much higher extent upon choline bitartrate than upon Superba*Boost*^TM^. No TMAO increase was observed in the fish oil group (*p* = 0.1633).

Pharmacokinetic parameters of TMAO upon intake of the study and control products are shown in Table 5. Of note, AUC_0–12 h_, AUC_0–24 h_ and C_max_ levels were significantly higher upon intake of choline bitartrate compared to Superba*Boost*^TM^. T_max_ levels, on the other hand, were comparable between both choline sources.

### 3.4. Safety Assessment

The tolerability of both study products was very good and only minor reports such as a fishy aftertaste after capsule intake were documented. No clinically relevant findings were documented during the course of the study. No serious adverse events (SAEs) occurred. During the kinetic days, adverse events not related to study product intake were reported by two subjects, one of whom reported a headache on both kinetic days. The other subject reported headache on the kinetic day after Superba*Boost*^TM^ intake.

## 4. Discussion

In the present study, the absorption pattern and pharmacokinetic pattern of free choline from Superba*Boost*^TM^ krill oil and choline bitartrate were investigated. In Superba*Boost*^TM^, choline is bound in a lipid-soluble form as phosphatidylcholine, while it is present as a water-soluble salt structure in choline bitartrate. Results show that free choline levels in plasma were comparable between Superba*Boost*^TM^ and choline bitartrate, but peak choline concentrations were reached significantly later upon intake of Superba*Boost*^TM^ compared to choline bitartrate. Moreover, levels of the choline metabolites betaine and DMG were higher, while levels of TMAO were lower upon intake of Superba*Boost*^TM^ compared to choline bitartrate.

The dependency between choline intake and circulating levels of free choline and its metabolites is concealed by homeostatic regulations and rapid tissue uptake [46,47]. Thus, concentrations of choline and its metabolites are usually within a narrow range. In the literature, endogenous free fasting choline levels under physiological conditions range between 7–12 µmol/L [2,44,47,48], which could be confirmed in the present study with a mean choline concentration of 9.6 µmol/L before product intake. Moreover, baseline values were highly comparable within all groups and within the two kinetic days.

It is well known that choline levels are influenced by nutrition, and choline sources depend on the diet [6], In a normal Western diet, choline can be obtained from eggs, liver, soybeans and pork [6,18], where choline is present primarily as phosphatidylcholine. It was shown that the consumption of a small breakfast led to 25–30% higher choline levels compared to the fasting condition [45], and that a high choline-containing diet resulted in an up to two-fold increase in plasma choline [49]. In a recent study by Cho et al., eggs were provided as a choline source, which contained comparable amounts of choline (479 mg) as the herein investigated study products (choline bitartrate: 620 mg; Superba*Boost*^TM^: 572 mg). Cho et al. reported a 1.6-fold increase in free choline after consumption of the egg meal, in comparison to the fruit control meal [47]. Correspondingly, the choline increase in the current study upon supplement intake was comparable (choline bitartrate: 1.35-fold; Superba*Boost*^TM^: 1.48-fold).

To control for confounding factors of food, subjects of the present study adhered to a diet low in choline three days before the study intervention and consumed standardized meals during the kinetic days. In addition, a parallel study arm was included, whose subjects underwent a single kinetic dose after the intake of fish oil as a placebo control that was devoid of choline. Of note, we observed a minor increase in choline levels in the placebo group after the 4 h time point; thus, the standardized lunch served after four hours could have contributed to this elevation. However, no such effect was seen in the study groups receiving choline supplements. In addition, by implementing the control group, we were able to exclude possible circadian rhythm-dependent variations in choline levels, in contrast to Park et al., who observed a slight elevation of free choline levels in the afternoon during a 12 h-lasting observation of endogenous choline levels [44].

Krill oil is increasingly recognized as a useful source of phosphatidylcholine [15,16], in addition to its acknowledged role in providing the omega-3 fatty acids EPA and DHA. In a former study, phosphatidylcholine was shown to raise plasma choline levels more efficiently compared to ingestion of free choline as choline chloride [50,51]. In the current study we observed a significant increase of choline after single doses of both phosphatidylcholine and the reference product choline bitartrate; however, phosphatidylcholine was not more efficient, since there were no significant differences in the pharmacokinetic endpoints AUC and C_max_. This discrepancy compared to the former study could be due to possible differences in the bioavailability of the chosen choline salts. However, to the best of our knowledge, choline bitartrate and choline chloride bioavailability has not been directly compared to date. Still, the uptake profile of free choline in the current study indicated a slower increase upon phosphatidylcholine (Superba*Boost*^TM^) intake compared to choline bitartrate. This difference in T_max_ between the study products could result from the structural differences between the choline forms and from the associated differences in absorption and metabolism. Phosphatidylcholine and/or its isoforms may be absorbed intact and can also be hydrolyzed into phosphatidic acid and free choline by phospholipase D [17]. Due to this conversion step, it supposedly takes longer to obtain free choline from phosphatidylcholine compared to choline bitartrate, where free choline from the salt structure is more rapidly available. Our findings are in agreement with former findings showing that choline levels after choline chloride reached peak levels earlier and were reduced to baseline earlier than after phosphatidylcholine intake, where levels remained elevated for 12 h [50].

After intake, choline is extensively transformed into different metabolites, such as betaine. Betaine is an oxidation product of free choline and serves as substrate in the betaine–homocysteine methyltransferase reaction, which links choline and betaine to folate-dependent one-carbon metabolism. Thus, choline and folate levels are closely interconnected, and deficiencies in either of these compounds might aggravate the nutrition status of the counterpart [52,53,54,55]. Therefore, betaine is an important choline metabolite, but is also present itself in various foods such as cereals, spinach, quinoa or noodles. Betaine can be further metabolized to DMG, which is a crucial methyl group donor for many biochemical reactions in the body. Both metabolites are usually present in plasma prior to a choline challenge from supplements. Respective mean values in plasma of different healthy cohorts are reported to range between 30–39 µmol/L (betaine) and 2.5–4.0 µmol/L (DMG) [45,48,56]. Comparable betaine and DMG levels were measured in the current study cohort under baseline conditions (Table 1). Substantial amounts of the ingested choline were converted into betaine and DMG, raising the pre-challenge levels both upon choline bitartrate intake and upon phosphatidylcholine intake from krill oil. Similar results were presented in a recent study by Bjørndal and colleagues, showing that krill oil supplementation significantly induced fasting plasma levels of choline, betaine and DMG [57]. As a study limitation, the authors named the missing placebo control devoid in phosphatidylcholine but containing omega-3 fatty acids in similar amounts as the krill oil product. This control group was included in the present study, confirming that the pharmacokinetic effects were attributable to phosphatidylcholine only, without confounding influences by omega-3 fatty acids.

The extent of the betaine (choline bitartrate: 1.5-fold; Superba*Boost*^TM^: 1.88-fold) and DMG (choline bitartrate: 1.31-fold; Superba*Boost*^TM^: 1.35-fold) increase observed in the present study was comparable to earlier results by Cho et al., with 1.2-fold and 1.3-fold increases, respectively [47]. Of note, we demonstrated differences in the pharmacokinetic parameters AUC_0–12 h_ and C_max_ of betaine, which were higher after intake of Superba*Boost*^TM^ compared to choline bitartrate. Also, the AUC_0-12 h_ of DMG was slightly higher upon Superba*Boost*^TM^ compared to choline bitartrate. However, these differences in both betaine and DMG were not confirmed for AUC_0-24 h_. Again, a delayed conversion from phosphatidylcholine to free choline could have contributed to higher and longer maintained levels of choline metabolites in the circulation in comparison to a faster choline metabolism upon choline bitartrate intake. Moreover, plasma phosphatidylcholine levels after Superba*Boost*^TM^ intake were not measured in the current study, but could have served as some kind of depot from which further betaine and DMG was generated, in contrast to choline bitartrate.

On the other hand, possible differences between the study products in regard to the T_max_ of betaine and DMG might have been missed due to the limited sampling times between 4 and 12 hours after dosing. Of note, a slight elevation in betaine levels after breakfast, and a more substantial elevation after lunch, was observed in the fish oil control group, which could have been due to the betaine content in the served oatmeal and noodles, respectively. Thus, similar lunch-mediated effects on betaine levels in the study groups cannot be excluded. Yet, C_max_ levels were significantly elevated in the study groups compared to the control group (choline bitartrate: *p* = 0.0042; Superba*Boost*^TM^: *p* = 0.0002).

The most prominent difference observed between phosphatidylcholine and choline bitartrate was observed in regard to TMAO plasma levels after intake of the study products. Substantially higher TMAO concentrations were measured after intake of the choline bitartrate salt, compared to phosphatidylcholine from Superba*Boost*^TM^. This finding was recently confirmed in healthy young adults, where krill oil supplementation resulted in increased choline and betaine but not TMAO levels [58]. TMAO is a metabolite of TMA, which is generated primarily by the action of gut microbiota from dietary choline, betaine, l-carnitine and its metabolite y-butyrobetaine [21,22,23,24]. In addition, TMAO is a major osmolyte in fish and other seafoods and is detectable in low levels in healthy people, as also demonstrated in this study. A recent meta-analysis of 11 studies demonstrated that elevated TMAO levels, higher than levels ranging from 5 to 70 µmol/L, were associated with the risk of cardiovascular disease [59]. The fasting concentrations of TMAO in our study (mean TMAO: 2.1 µmol/L) were below this range, and currently there are no studies to link short-termed elevated TMAO levels after a choline bolus to cardiovascular risks.

## 5. Conclusions

The objective of the current study was to compare the plasma kinetics of choline and its turnover into choline derivatives upon intake of two different choline sources: phosphatidylcholine from krill oil (Superba*Boost*^TM^) and choline bitartrate. Although the plasma concentration vs. time profiles of free choline itself were comparable between both choline sources, a distinct rate of choline metabolism to its derivatives (betaine, DMG and TMAO) was observed. While betaine and DMG levels were somewhat elevated following phosphatidylcholine intake compared to choline bitartrate, the inverse pattern was observed regarding TMAO plasma levels, which were higher upon choline bitartrate intake than phosphatidylcholine. These data clearly indicate that choline metabolism by the gut microbiome, and thus TMAO generation, varies considerably depending on the provided choline source. Besides being an acknowledged source for omega-3 fatty acids [60], this study clearly confirmed that Superba*Boost*^TM^ krill oil is also an adequate source for choline and for its metabolites betaine and DMG, without elevating TMAO levels.

## Figures and Tables

**Figure 1 nutrients-11-02548-f001:**
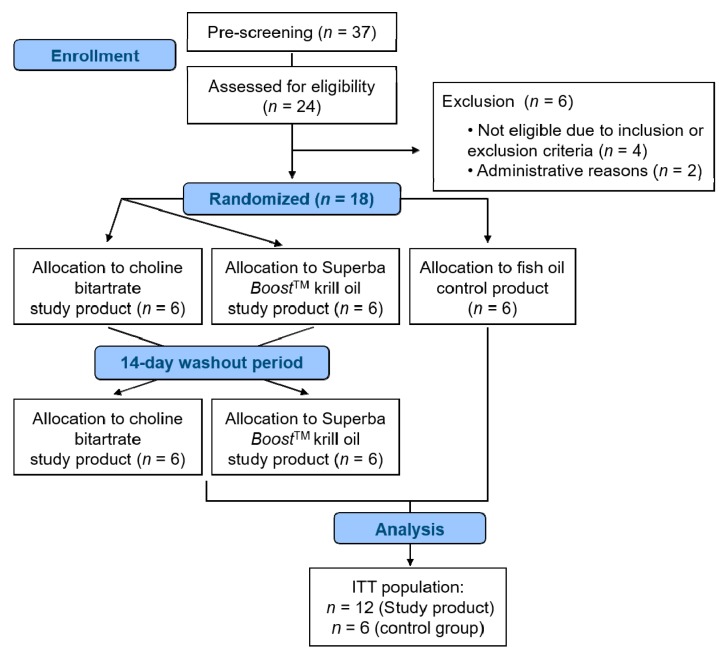
Flow chart of subject recruitment and included study subjects. ITT: intention-to-treat.

**Figure 2 nutrients-11-02548-f002:**
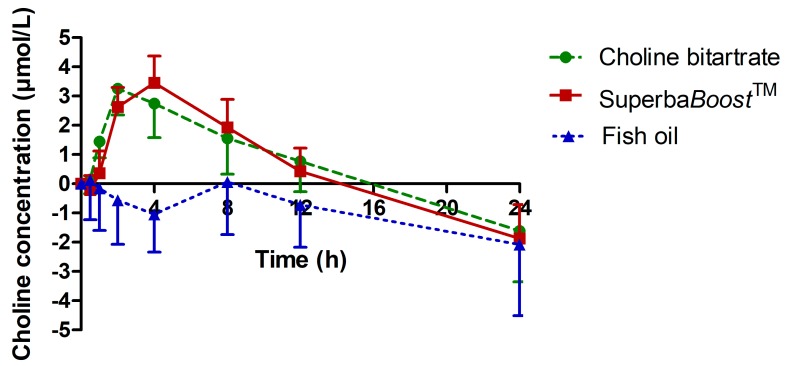
Plasma concentration–time curve of choline. Values are baseline-corrected and dose-adjusted. Choline bitartrate: *n* = 12; Superba*Boost*^TM^: *n* = 12; fish oil: *n* = 6.

**Figure 3 nutrients-11-02548-f003:**
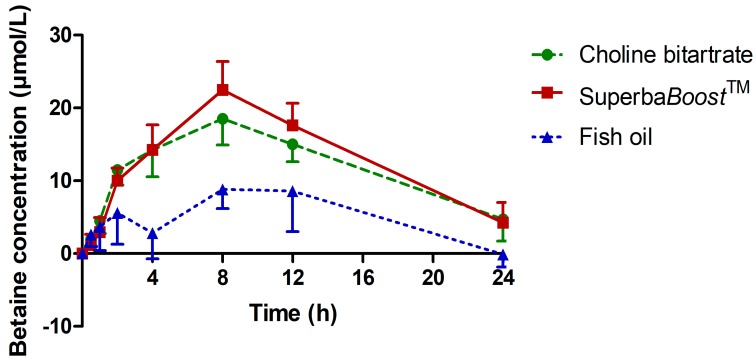
Plasma concentration–time curve of betaine. Values are baseline-corrected. Choline bitartrate: *n* = 12; Superba*Boost*^TM^: *n* = 12; fish oil: *n* = 6.

**Figure 4 nutrients-11-02548-f004:**
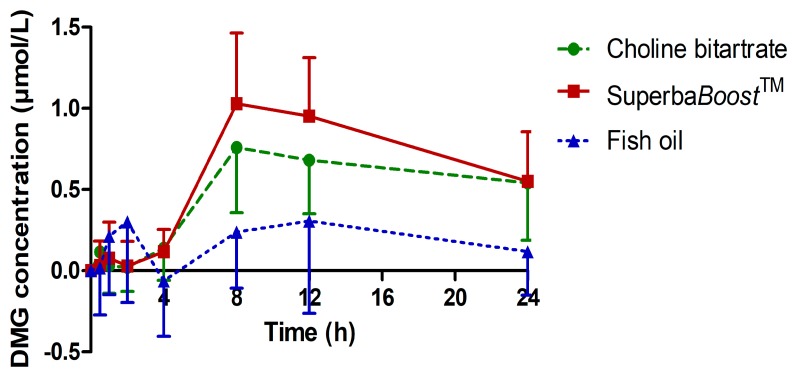
Plasma concentration–time curve of DMG. Values are baseline-corrected. Choline bitartrate: *n* = 12; *Superba*Boost^TM^: *n* = 12; fish oil: *n* = 6.

**Figure 5 nutrients-11-02548-f005:**
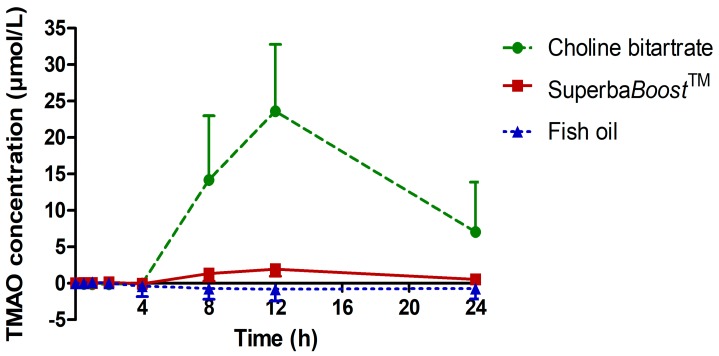
Plasma concentration–time curve of TMAO. Values are baseline-corrected. Choline bitartrate: *n* = 12; Superba*Boost*^TM^: *n* = 12; fish oil: *n* = 6.

**Table 1 nutrients-11-02548-t001:** Demographic and baseline data of the study cohort.

Variable	Mean	95% CI
Age (years)	43.3	34.0–52.6
BMI (kg/m^2^)	23.6	21.8–25.3
Systolic BP (mmHg)	128.5	117.6–139.4
Diastolic BP (mmHg)	78.5	72.6–84.4
Hemoglobin (g/dL)	14.2	13.4–14.9
Cholesterol (mg/dL)	180.5	155.1–205.9
HDL (mg/dL)	60	51.6–68 4
LDL (mg/dL)	112 2	86.3–138
AST (U/L)	21.6	19.6–23.6
ALT (U/L)	24.2	20.3–28
Choline (baseline) (µmol/L)	9.6	8.5–10.6
Betaine (baseline) (µmol/L)	33.8	28.5–39.1
DMG (baseline) (µmol/L)	3.4	3–3.8
TMAO (baseline) (µmol/L)	2.1	1.4–2.8

Abbreviations: CI: confidence interval; BP: blood pressure; BMI: body mass index; HDL: high-density lipoprotein; LDL: low-density lipoprotein; AST: aspartate aminotransferase; ALT: alanine aminotransferase; DMG: dimethylglycine; TMAO: trimethylamine N-oxide.

**Table 2 nutrients-11-02548-t002:** Pharmacokinetic parameters of choline.

Variable	Mean ^1^	95% CI ^1^	*p*-Value ^2^
AUC_0–12 h_ ((µmol/L)*h) (SB)	26.07	19.72–32.42	0.2966
AUC_0–12 h_ ((µmol/L)*h) (CB)	23.28	13.61–32.95	
AUC_0–24 h_ ((µmol/L)*h) (SB)	29.57	21.26–37.87	0.4144
AUC_0–24 h_ ((µmol/L)*h) (CB)	30.36	12.88–47.85	
C_max_ (µmol/L) (SB)	4.21	3.43–4.99	0.2690
C_max_ (µmol/L) (CB)	3.67	2.82–4.51	
T_max_ (h) (SB)	4.17	2.9–5.43	0.0076
T_max_ (h) (CB)	2.67	2.04–3.29	

Abbreviations: SB: Superba*Boost*^TM^; CB: choline bitartrate; CI: confidence interval; AUC: area under the curve; C_max_: peak concentration; T_max_: time to reach maximum concentration. ^1^ Dose-adjusted values; ^2^ Difference between study products SB and CB.

**Table 3 nutrients-11-02548-t003:** Pharmacokinetic parameters of betaine.

Variable	Mean	95% CI	*p*-Value ^1^
AUC_0–12 h_ ((µmol/L)*h) (SB)	185.8	154–217.6	0.0641
AUC_0–12 h_ ((µmol/L)*h) (CB)	168.4	137.4–199.4	
AUC_0–24 h_ ((µmol/L)*h) (SB)	317.1	255.3–378.9	0.2394
AUC_0–24 h_ ((µmol/L)*h) (CB)	286.9	229.0–344.8	
C_max_ (µmol/L) (SB)	22.55	18.62–26.48	0.0122
C_max_ (µmol/L) (CB)	18.87	15.31–22.42	
T_max_ (h) (SB)	7.67	6.93–8.40	1.0
T_max_ (h) (CB)	7.67	6.36–8.98	

^1^ Difference between study products SB and CB.

**Table 4 nutrients-11-02548-t004:** Pharmacokinetic parameters of DMG.

Variable	Mean	95% CI	*p*-Value ^1^
AUC_0–12 h_ ((µmol/L)*h) (SB)	6.75	3.99–9.51	0.0890
AUC_0–12 h_ ((µmol/L)*h) (CB)	5.17	2.55–7.78	
AUC_0–24 h_ ((µmol/L)*h) (SB)	15.79	10.11–21.46	0.1546
AUC_0–24 h_ ((µmol/L)*h) (CB)	12.74	7.13–18.35	
C_max_ (µmol/L) (SB)	1.22	0.81–1.61	0.1614
C_max_ (µmol/L) (CB)	0.95	0.64–1.27	
T_max_ (h) (SB)	12.0	8.25–15.75	0.9336
T_max_ (h) (CB)	13.33	8.21–18.45	

^1^ Difference between study products SB and CB.

**Table 5 nutrients-11-02548-t005:** Pharmacokinetic parameters of TMAO.

Variable	Mean	95% CI	*p*-Value ^1^
AUC_0–12 h_ ((µmol/L)*h) (SB)	10.06	4.19–15.92	<0.0001
AUC_0–12 h_ ((µmol/L)*h) (CB)	104.3	59.8–148.8	
AUC_0–24 h_ ((µmol/L)*h) (SB)	25.41	11.76–39.06	<0.0001
AUC_0–24 h_ ((µmol/L)*h) (CB)	288.3	231.8–362.8	
C_max_ (µmol/L) (SB)	2.24	1.17–3.30	<0.0001
C_max_ (µmol/L) (CB)	28.99	20.36–37.63	
T_max_ (h) (SB)	12.0	9.35–14.65	0.7884
T_max_ (h) (CB)	11.67	8.91–14.42	

^1^ Difference between study products SB and CB.

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
