# Peer review of "Plasma Kinetics of Choline and Choline Metabolites After A Single Dose of SuperbaBoostTM Krill Oil or Choline Bitartrate in Healthy Volunteers"

_nutrients, 2019, doi:10.3390/nu11102548_

Round 1

Reviewer 1 Report

This is a well-thought out study that accomplished the goal proposed: to test the ability of Krill oil to increase plasma choline concentrations. The authors also appropriately measured plasma concentrations of choline derived metabolites and compare results to a choline-less fish oil, as well as a pure choline supplement. As such this is a worthy manuscript that fits within the scope of the journal. Some minor english editing is required, and I suggest a native englishspeaker look it over to catch some awkward sentences and choice of diction. However, despite these stylistic limitations, the manuscript can be well understood.

I have two main issues to point out. I believe that that the value of the manuscript could be improved by adding more justifications for the study in the introduction, and by making grounding the conclusions more within the literature, particularly among related studies.

In the case of the introduction I would like to see the following questions answered and the following corrections made.

1) Line 58: substitute "unprecedented" for "apparently harmful".

2) With that in mind, what levels of TMAO are considered harmful?

3) Also, perhaps here it should be emphasized that any choline supplement should be tested for its effects TMAO levels to prevent possible harmful side effects and that generally it may be a good idea to limit the introduction of TMAO via a supplement. This is important, because there are dramatic differences in TMAO levels with the Krill oil vs the choline, therefore it could be argued that the Krill oil (and phosphotidylcholine generally) has potentially fewer health deficits compared to pure choline, which I thing is a strong point to make in favor of the Krill oil.

3) Why did you choose Supraboost Krill oil over other sources of choline or phosphoatydyl choline (PC))? How did you expect it to differ, or be comparable sources already tested in the literature? Is it superior to other sources because it is cheaper to produce, easier to deliver, has more consumer approval, better characterized etc?

4) I think it makes more sense to introduce the pathway for how choline is liberated from PC in the introduction, to expain better why such a source of choline is viable. A related question would be, is it your expectation that choline should appear later in the plasma with Krill oil because it must first be liberated from PC.

5) 90% of people are not meeting their choline intake requirements: which is how much? compared to how much in the krill oil of choline samples?

6) What is a healthy baseline of choline in plasma? (maybe discuss in the conclusion rather than introduction).

In the conclusion:

The results with each metabolite should be compared with the following:

What are the healthy values of each

Reviewer 2 Report

This is a cohort study to compare metabolites in healthy adult between SuperbaBoost krill oil and choline bitartrate. The results indicated the major difference between the two choline sources is the level of Trimethylamine N-oxide (TMAO), which might associate with an increased risk of major adverse events in humans. There are a few major concern need to be address.

Authors should include the detailed anthropometric and laboratory parameters of the volunteers for inclusion and exclusion conditions. At least including the basic plasma phosphatidylcholine or choline, but also not limited in Apo-B, Hemoglobin, Cholesterol et. Al。 The “phosphatidylcholine is only limitedly converted to the undesirable trimethylamines”. It seems like the major result of this study a redundant repeat. The data in this study seems not quite convince to support the conclusion, although the conclusion is just “might have an advantage”

Round 2

Reviewer 2 Report

I thank authors response and am satisfied with that. 

Author Response

We thank the reviewer again for the valuable remarks, and are content that all issues were sufficiently taken into account during the last review circle.